# Key Role of Mesenchymal Stromal Cell Interaction with Macrophages in Promoting Repair of Lung Injury

**DOI:** 10.3390/ijms24043376

**Published:** 2023-02-08

**Authors:** Mirjana Jerkic, Katalin Szaszi, John G. Laffey, Ori Rotstein, Haibo Zhang

**Affiliations:** 1The Keenan Research Centre for Biomedical Science of St. Michael’s Hospital, Unity Health Toronto, University of Toronto, Toronto, ON M5B 1T8, Canada; 2Department of Surgery, University of Toronto, Toronto, ON M5T 1P5, Canada; 3Anaesthesia and Intensive Care Medicine, School of Medicine, University of Galway, H91 TK33 Galway, Ireland; 4Department of Anesthesiology and Pain Medicine, Interdepartmental Division of Critical Care Medicine and Department of Physiology, University of Toronto, Toronto, ON M5G 1E2, Canada

**Keywords:** acute and chronic respiratory diseases, cell therapy, lung immunity and inflammation

## Abstract

Lung macrophages (Mφs) are essential for pulmonary innate immunity and host defense due to their dynamic polarization and phenotype shifts. Mesenchymal stromal cells (MSCs) have secretory, immunomodulatory, and tissue-reparative properties and have shown promise in acute and chronic inflammatory lung diseases and in COVID-19. Many beneficial effects of MSCs are mediated through their interaction with resident alveolar and pulmonary interstitial Mφs. Bidirectional MSC-Mφ communication is achieved through direct contact, soluble factor secretion/activation, and organelle transfer. The lung microenvironment facilitates MSC secretion of factors that result in Mφ polarization towards an immunosuppressive M2-like phenotype for the restoration of tissue homeostasis. M2-like Mφ in turn can affect the MSC immune regulatory function in MSC engraftment and tissue reparatory effects. This review article highlights the mechanisms of crosstalk between MSCs and Mφs and the potential role of their interaction in lung repair in inflammatory lung diseases.

## 1. Introduction

Mesenchymal stromal cells (MSCs) are now known to enhance the repair of injured tissues and are emerging as possible therapeutic agents in acute and chronic inflammatory lung diseases and in COVID-19. The interaction between MSCs and Mφs has been shown to be a crucial mechanism of such beneficial action in lung injury. Our review focuses on this fascinating crosstalk. We describe the current understanding of the mechanisms and complexity of the MSC-Mφ interplay. We also point out the numerous gaps in our knowledge that still hinder exploiting this mechanism for therapeutic purposes. The final outcome of MSC-Mφ interaction depends on multiple factors, and a better understanding of these is essential for designing strategies to improve the efficacy of MSC treatment and maximize positive and avoid possible negative outcomes. We also provide an overview of the potential relevance of the MSC-Mφ interplay, as emerging from preclinical and clinical studies. Since the therapeutic efficacy of MSCs is still clinically unproven, this knowledge is crucial for personalized therapy for lung injury patients.

## 2. The Role of Lung Tissue Macrophages in the Pathogenesis of Lung Injury

### 2.1. Pulmonary Mφs as a Host Defense Mechanism

The lung is not only vital for gas exchange but also serves as a major immune organ that protects the host from inhaled pathogens, allergens, and toxins [1]. Since pulmonary Mφs are fundamental for the regulation of tissue homeostasis, modulation of their function could be used to prevent injury and promote repair both in acute and chronic lung injury.

Both alveolar (AMφs) and pulmonary interstitial (PIntMφs) Mφs are essential components of lung innate immunity and the host defense system [2]. These Mφs are originated from fetal erythro-myeloid progenitors (yolk-sac-derived) [3,4], although there is some age-dependent contribution of circulating adult monocytes to their pool [5,6].

AMφs represent the first line of host defense. They are defined by expression of MHCII (HLA-DR), CD11c, macrophage receptor with collagenous structure (MARCO), granulocyte–macrophage colony-stimulating factor receptor (GM-CSFR), CD206, and CD169.

In the absence of infection, AMφs primarily regulate the secretion of pulmonary surfactant, i.e., the lipid–protein complex produced by alveolar type II (ATII) cells. They are essential for lung homeostasis, as spontaneous pulmonary alveolar proteinosis (PAP) develops in mice and humans with absent or dysfunctional AMφs [7,8,9].

AMφs eliminate the small load of typical microbes aspirated daily in the normal host via phagocytosis and intracellular killing and by secretion of reactive oxygen species (ROS), antimicrobial peptides, proteases, and lysozyme [10]. Macrophage functional defects inevitably lead to increased susceptibility to a variety of bacteria, viruses, and fungi [11].

PIntMφs mainly express MHCII, CD11b, Lyve-1, and/or CD36 [12] and act as a second line of defense should the epithelial barrier be damaged. They constitutively secrete interleukin-10 (IL-10), suggesting an immunoregulatory role in both mouse and man [13,14,15,16]. They also produce platelet-derived growth factor (PDGF), suggesting they may support fibroblast, epithelial, and endothelial homeostasis [17].

### 2.2. Role of Mφ Plasticity in Lung Injury and Repair

During acute inflammation, damage-associated molecular patterns (DAMPs), cytokines, and growth factors (GM-CSF and M-CSF) [18] induce the recruitment and differentiation of residual yolk-sac-derived pulmonary Mφs and promote their interaction with surrounding cells. Bone-marrow-derived monocytes are also recruited to the lung and differentiate into AMφs upon inflammatory stimuli [19]. Cytokines and chemokines, especially CCL2 (also referred to as monocyte chemoattractant protein 1—MCP1) and its receptor CCR2, play a crucial role in Mφ recruitment and polarization [20].

The activation and polarization of Mφs into two extreme phenotypes, M1 (pro-inflammatory) and M2 (involved in the resolution of inflammation and tissue repair), was described in the early 1990s [21]. M1-like Mφs start and sustain inflammatory responses, while M2-like Mφs promote the resolution of inflammation and coordinate the restoration of tissue integrity (Figure 1).

Mφ balance is swayed towards an M1-like state by microbial products (lipopolysaccharide (LPS) and other Toll-like receptor (TLR) ligands), and the pro-inflammatory microenvironment activates nuclear factor kappa-light-chain-enhancer of activated B cells (NF-κB) and signal transducer and activator of transcription-1 (STAT-1). In addition, Th-1 lymphocytes secrete interleukins (IL)-1β, IL-6, IL-17, IL-18, IL-23, interferon gamma (IFNγ), and tumor necrosis factor (TNF)-α, and activated platelets further promote M1 polarization (Figure 1) [22,23,24]. Oxidative stress and ROS increase also contribute to tissue damage [25]. In contrast, Th-2 cytokines (e.g., IL-4 and IL-13) induce M2-like or alternative Mφ activation [26]. Different stimuli induce various subtypes of M2-like Mφs (M2a, b, c, and d) (detailed in an excellent review by Wang et al., 2019 [27]).

Importantly, Mφ responses to environmental challenges are complex, and their functional polarization into M1 and M2 types is an over-simplification. Rather than being an “on-off” process, current data clearly indicate that Mφ polarization is a continuum along the whole spectrum between M1 and M2 phenotypes. Therefore, terms such as M1-like and M2-like Mφs more appropriately describe the phenotypes [28].

Of note, neither AMφs nor PIntMφs can be defined exclusively by M1 or M2 markers, and in the healthy lung both populations co-express markers historically considered as M1- or M2-specific [12,29,30].

The above-described plasticity renders Mφs attractive candidates for therapeutic purposes, as environmental influences can dynamically and reversibly alter their phenotype [23,28]. For example, our group has successfully used pluripotent stem cell-derived Mφs in rats to treat systemic sepsis with multi-organ involvement that includes lung injury [31]. Indeed, Mφ transplantation therapy may ultimately be used as a therapeutic approach for lung injury, but its feasibility and effects in humans have yet to be investigated. So far, Mφs have not typically been considered as cells that can be directly administered to patients with respiratory diseases, in part because they would likely elicit an immune response. Rather, therapeutic strategies mainly focus on harnessing the plasticity of existing pulmonary Mφs.

In this respect, mesenchymal stromal cell (MSC) treatment offers an opportunity to interact with host macrophage populations and modulate their function and vice versa, as described in the next sections.

Lung tissue macrophages (Mφs), alveolar (AMφs) and interstitial (PIntMφs), are essential for host defense. AMφs eliminate the small daily load of aspirated microbes and regulate pulmonary surfactant. PIntMφs have anti-inflammatory and regulatory roles [1,2].

Inflammation attracts circulating Mφs (by CCL-2) [20] and causes Mφ proliferation (induced by GM-CSF) [18,19]. Pathogen sensors, TLR-4, and other TLRs cause NF-κB and STAT-1 pathway activation and secretion of pro-inflammatory interleukins. This environment shifts Mφ balance towards an M1-like state and induces ROS production causing tissue damage and edema [22,23,24,25]. Conversely, the regeneration and healing process is mainly driven by alternatively activated (M2-like) Mφs, with the presence of IL-4, IL-10, TGF-β [32], and antioxidants (Nrf2/HO-1 axis) [33,34]. Mφ polarization is reversible and M1/M2 balance may vary depending on the local environment [27,35].

## 3. MSCs in Lung Injury Treatment

### 3.1. Definition and Characteristics of MSCs

MSCs are multipotent cells capable of differentiating into at least three lineages: adipocyte, chondrocyte, and osteoblast. These plastic-adherent cells have secretory, immunomodulatory, and homing properties. They express CD73, CD90, and CD105 and lack the expression of hematopoietic and endothelial markers CD11b, CD14, CD19, CD34, CD45, CD79a, and HLA-DR (ISCT’s MSC committee criteria [36]).

MSCs form a supportive, perivascular niche for hematopoietic stem cells (HSCs) in the bone marrow and coordinate the trafficking of HSCs and monocytes [37,38]. Although initially identified in bone marrow (BM) [39], MSCs are also present as perivascular cells in other tissues, including muscle, umbilical cord, and adipose tissue [40].

### 3.2. Preclinical Studies—MSC Action, Licensing, and Genetic Modifications

Acute respiratory distress syndrome (ARDS) is the leading cause of morbidity and mortality (30–50%) in the critically ill receiving supportive treatment. Despite some improvements, there is still no causal therapy available [41,42]. The hallmarks of ARDS include alveolar epithelial–capillary barrier disruption, consequent edema formation, and widespread uncontrolled lung inflammation. In preclinical studies (Table 1), the treatment of ARDS rodent models with MSCs [43] or with MSC-derived extracellular vesicles (EVs) significantly mitigated lung injury [44]. Interestingly, EVs isolated from young MSCs were more effective [45]. Studies also demonstrated that MSCs or MSC-EVs acted by decreasing the production of pro-inflammatory cytokines by AMφs [44] and/or by inducing alveolar–endothelial barrier restoration partially via mitochondrial transfer [46]. Administration of MSCs protected rodents from ventilator-induced lung injury [47,48], with MSCs being equally [47] or more effective than their secretome [49]. MSCs were also found to be beneficial in mice with sepsis-induced lung injury. In these models, the improvement was ascribed to the downregulation of miR-27a-5p microRNA [50] and microRNA (miR)-193b-5p [51] in the septic lung and upregulation of their respective target genes, i.e., VAV3 and the tight junctional protein occludin.

Several studies demonstrated that modification of MSCs can augment their beneficial effects. The immune-modulatory and tissue-reparatory properties of MSCs can be enhanced by pre-stimulation, i.e., licensing or by their transfection with targeted genes prior to the treatment. Cytokine-induced pre-activation (with IL-1β, TNF-α, IFN-γ) augmented MSC-induced repair and resolution of ventilator-induced lung injury [52]. EVs from interferon (IFN)-γ-primed human umbilical cord (hUC)-MSCs were more beneficial in *E. coli*-induced lung injury in rats than EVs from naïve MSCs [53]. MSCs cultured in hypoxic conditions displayed more protection in radiation-induced lung injury in mice by promoting MSC viability and improving their antioxidant capacity [54]. A hypoxic environment promoted EV release by MSCs and enhanced their potency in suppressing airway inflammation in asthmatic mice [55]. Hyperthermia increased the efficacy of MSC-driven immune suppression [56].

Genetic modification of MSCs can be achieved by overexpression or silencing of specific genes using different knock-in and knock-out technologies including CRISPR (clustered regularly interspaced short palindromic repeats)/Cas9 gene-editing system, RNA interference technology, etc. These manipulations could be used to control native MSC gene expression or introduce foreign genes for specific therapeutic applications (as reviewed in detail in Varkouhi et al., 2020 [57]).

Our group has shown that overexpression of IL-10 enhanced the efficacy of hUC-MSCs in *E. coli* pneumosepsis in rats [58]. However, of note, in acid-primed lung injury associated with the development of fibrosis, MSC treatment was harmful. Correction of the microenvironment after acid-primed lung injury or treatment with MSCs carrying the human IL-10 gene or hepatocyte growth factor (HGF) reversed the detrimental effects of naïve MSCs [48]. Therefore, the lung microenvironment, disease type, and severity have to be taken into account as they affect MSC activation, function, and therapeutic effectiveness in both acute and chronic lung diseases [59,60].

MSCs have been proven effective in various models of chronic respiratory diseases too, including Chronic Obstructive Pulmonary Disease (COPD/emphysema) [61,62,63], asthma [55], and bronchopulmonary dysplasia (BPD) [64]. Tang et al. [65] found that human hUC-MSCs could attenuate bleomycin-induced pulmonary fibrosis in mice by acting on Mφs, reducing CD206 Mφ number, and recruiting T-regulatory cells.

**Table 1 ijms-24-03376-t001:** Preclinical studies (in vitro and in vivo) of MSC treatment in lung conditions.

Preclinical Model	Intervention (MSCs/EVs)	Outcome	Mechanism	References
Human-monocyte-derived Mφs in noncontact co-culture with hMSCs.	Stimulation of co-cultured cells with LPS or BALF from patients with ARDS.	MSCs suppressed pro-inflammatory cytokine production by Mφ.	Increased M2 Mφ marker expression and augmented phagocytic capacity of Mφs.	[44] Morrison et al., 2017
MSCs cultured under different temperatures in vitro in co-culture with Mφ.	hBM-MSCs and Mφ.	MSCs cultured at higher temperatures induce more IL-10 and less TNFα production in Mφs (M2-like phenotype).	Nuclear translocation of HSF-1 and induction of COX2/PGE2 pathways by hyperthermia in MSCs promoted M2-like Mφ phenotype change.	[56] McClain-Caldwell et al., 2018
Polymicrobial sepsis-induced lung injury in mice and in vitro.	Murine MSCs or MSC-conditioned media.	Attenuation of sepsis and TNF-induced miR- 193b-5p upregulation.	miR-193b-5p was decreased by MSCs while its target gene OCLN was increased in lungs from septic mice and in vitro.	[51] DosSantos et al., 2022
*Escherichia coli* (*E. coli*)-induced ARDS in rats.	hUC-MSCs and hBM-MSCs.	Improved animal survival, systemic oxygenation, and lung compliance by both hUC- and BM-MSCs.	Decrease in pro-inflammatory cytokines in BALF, increase in IL-10, and ROS reduction in lung tissue.	[43] Curley et al., 2017
LPS-induced ALI in mice.	Adoptive transfer of AMφs pretreated with hMSC-derived EVs.	Reduced inflammation and lung injury in LPS mice.	Mφ changes induced by mitochondrial transfer from EVs to AMφs during pretreatment.	[44] Morrison et al., 2017
LPS-induced ALI in mice.	MSC-EVs derived from young and aging MSCs.	Young MSC-EVs alleviated LPS-ALI, while aging MSC-EVs did not.	Aging MSC-EVs failed to be internalized and did not induce Mφ phenotypic change.	[45] Huang et al., 2019
LPS-induced ALI in mice.	MSC-EVs.	EVs reduce lung injury.	Restoration of mitochondrial respiration in the lung tissue.	[46] Dutra Silva 2021
Ventilator-induced ALI in rats.	Rodent BM-MSCs or their secretome.	Restored systemic oxygenation, lung function, and structure by both MSCs and their secretome.	Decreased lung inflammation (TNFα, IL-6), and increase in IL-10; role of KGF in lung repair.	[47] Curley et al., 2012
Ventilator-induced ALI in mice.	Murine BM-MSCs.	Lungs were protected from injury.	Improved lung function and reduced oxidative stress and collagen-1 expression.	[48] Islam et al., 2019
Ventilator-induced ALI in rats.	Rodent BM-MSCs or their secretome.	MSCs were more effective in reducing lung injury than their secretome.	Improved oxygenation; reduction in lung edema, alveolar inflammation, and IL-6 levels.	[49] Hayes et al., 2015
Polymicrobial sepsis-induced lung injury in mice.	Murine MSCs.	MicroRNA (miRNA) and transcriptome analysis of septic mouse lungs showed that MSCs induced a shift in transcription profiles favoring reconstitution of ‘sham-like’ expression patterns.	MSCs downregulated miR-27a-5p and upregulated its target gene VAV3 in septic lungs.	[50] Younes et al., 2020
Ventilator-induced ALI in rats.	hBM-MSCs, naïve and cytokine-pre-activated (with IL-1β, TNF-α, IFN-γ).	Cytokine pre-activation enhanced the capacity of MSCs to promote injury resolution.	Mechanism dependent on KGF secreted by MSCs.	[52] Horie et al., 2020
Radiation-induced pneumonia and late fibrosis in mice.	Murine BM-MSCs cultured in normoxic and hypoxic environment.	Therapeutic effect of MSCs exposed to hypoxia was more pronounced compared to MSCs exposed to normoxia.	Hypoxia-treated MSCs were more viable and resistant to hypoxia decreasing oxidative stress in lungs by HIF1-α.	[54] Li et al., 2017
Chronic asthma mouse model—challenged with ovalbumin (OVA).	hUC-MSCs-derived EVs from MSCs cultured in normoxic (Nor-EVs) and hypoxic (Hypo-EVs) conditions.	Hypo-EVs were more effective than Nor-EVs in attenuation of chronic asthma.	TGFβ1 pathway was decreased and miR-146-5p increased. The effect was more pronounced if Hypo-EVs were used.	[55] Dong et al., 2021
*E. coli*-induced pneumonia in rats.	EVs from naïve or interferon (IFN)-γ-primed hUC-MSCs.	EVs from IFN-γ-primed hUC-MSCs more effectively attenuated lung injury than EVs from naïve MSCs.	Enhancements of Mφ phagocytosis and bacterial killing.	[53] Varkouhi et al., 2019
*E. coli*-induced ARDS.	Naïve and IL-10 over-expressing hUC-MSCs.	IL-10-UC-MSCs were more efficient in decreasing structural lung injury compared to naïve UC-MSC or vehicle therapy.	AMφs from naïve and especially from IL-10-UC-MSC-treated rats enhanced Mφ phagocytosis via increased Mφ HO-1, an effect blocked by PGE2 and LXA4 inhibition.	[58] Jerkic et al., 2019
Acid-primed lung injury in mice.	Murine BM-MSCs, environment correction, or MSC-carrying human IL-10 or HGF gene.	MSCs worsened acid-primed lung injuries associated with fibrosis and high levels of ROS and IL-6.	Correction of oxidative stress with GPx-1, or treatment with MSCs carrying IL-10 or HGF after injury reversed the detrimental effects of naïve MSCs.	[48] Islam et al., 2019
COPD rat cigarette smoke model.	hUC-MSCs and hUC-EVs.	Both transplantation of hUC-MSCs and application of EVs reduced lung inflammation and ameliorated the loss of alveolar septa and their thickening.	Both hUC-MSCs and EVs decreased mononuclear infiltration and reduced the levels of NF-κB subunit p65 in COPD lungs.	[63] Ridzuan et al., 2021
Hyperoxia-induced bronchopulmonary dysplasia (BPD) in rats.	hUC-MSC-EVs.	EVs ameliorated the impaired alveolarization and pulmonary artery remodeling.	MSC-EV prevented hyperoxia-induced reduction in CD163-positive (M2-like) Mφ both in alveolar and interstitial compartment.	[64] Porzionato et al., 2021
Mouse-bleomycin-induced pulmonary fibrosis.	hUC-MSCs.	MSCs attenuated pulmonary fibrosis and promoted lung repair by interacting with Mφs.	Mφs interferon-sensitive sub-cluster induced by MSC infusion caused T-regulatory cell recruitment by CXCL9/10. Number of CD206 Mφs involved in fibrosis was reduced.	[65] Tang et al., 2021

Regardless of whether MSCs or their derivatives were used in the studies, their interplay with monocytes/macrophages is a key mechanism of therapeutic benefits [66,67]. As summarized in Table 1, most studies found Mφ involvement in the beneficial effects of MSCs in lung injury or described a role for alveolar IL-10 increase in the effect. It should be noted that IL-10 secretion mainly originates from Mφs or neutrophils and is involved in Mφ polarization towards an M2-like phenotype [68,69].

Co-culture experiments provide direct evidence for a role of MSC/Mφ interaction. MSCs were shown to enhance Mφ phagocytic [70,71] and anti-inflammatory capacity [72]. Moreover, MSC-derived exosomes inhibited M1 and promoted M2 polarization in LPS-stimulated Mφs [73], and MSCs primed with Mφ-derived conditioned media exhibited enhanced immunomodulatory potential [74]. Further evidence for a key role of the MSC/Mφ interaction was obtained in in vivo lung disease models. Specifically, systemic Mφ depletion experiments showed that both monocyte recruitment to the lungs and MSC/Mφ interaction are crucial for the favorable MSC effects in pulmonary diseases. Selective AMφs depletion reversed the therapeutic benefits of MSC treatment in a mouse model of allergic asthma [75] and in an *E. coli* ARDS mouse model [70]. Treatment of mice with BM-MSCs prevented the development of obliterative bronchiolitis after tracheal allografts, and this effect was eliminated by systemic depletion of Mφs [76]. Systematic depletion of Mφs also weakened the therapeutic effect of MSC-derived exosomes or MSCs in mouse models of severe asthma [73] and allergic airway inflammation [77], respectively.

While the general role of the MSC/Mφ interaction is now well documented, the details of this interplay remain poorly defined. A better understanding of this crucial interaction is essential since the therapeutic efficacy of MSCs in ARDS and other lung diseases remains unproven. More mechanistic insights into the MSC/pulmonary Mφ interaction will be needed for the optimization and personalization of MSCs in respiratory diseases.

## 4. Crosstalk between MSCs and Mφs—Mechanisms of Action

The MSC-Mφ crosstalk is now known to affect both cell types. Several mechanisms have been implicated as a vital mediator of these effects (see Figure 2). Most studies point out that administering live MSCs is much more effective therapeutically, although some studies have shown immunomodulatory effects and benefits of apoptotic MSCs instilled into septic animals [78,79]. Thus, improving MSC viability was seen as an important factor in augmenting cell efficacy in clinical trials for moderate to severe ARDS [80] precisely because the MSC-Mφ dyad is the key for the full spectrum of MSC action [66,81,82,83].

### 4.1. Contact-Dependent MSC-Mφ Interaction

#### 4.1.1. Receptor-Dependent Interaction

Many studies found a bidirectional effect of direct MSC-Mφ contact, affecting both cell types, that is important for the beneficial effects. Direct contact between MSCs and pro-inflammatory Mφs has been shown to reinforce tumor necrosis factor-stimulated gene-6 (TSG-6) production by MSCs, further promoting Mφs switching to an M2-like phenotype and suppressing T cell proliferation [84], and to lead to upregulation of CD200. Interaction with CD200R on Mφs in turn facilitated the reprogramming of Mφs towards an anti-inflammatory phenotype [84].

#### 4.1.2. Microtubular Network

Recent studies have shown that the transfer of MSC extracellular vesicles (EVs) and mitochondria (Mt) to Mφs could happen via a tunneling nanotubule formation that established direct MSC-Mφ contact [70,85].

Mitochondrial transfer to injured cells [86] or Mφs through nanotubes seems to be facilitated by mitochondrial Rho-GTPases [87,88] enhancing cell oxidative phosphorylation and Mφ phagocytosis in in vitro and in vivo models of ARDS [70,85,89].

### 4.2. MSCs and Mφ Secretome—Paracrine-Mediated Mechanisms

The majority of evidence to date supports the notion that the effects of MSCs on Mφs or donor cells are largely paracrine, i.e., through secreted factors [90].

The MSC/Mφ secretome (Table 2) consists of proteins (cytokines, chemokines, etc.), nucleic acids, lipids, and EVs that act as bioactive molecules involved in MSC/Mφ crosstalk. Moreover, these factors may mimic therapeutic effects of MSC transplantation [90,91].

#### 4.2.1. Role of Cytokines, the COX/PGE2/EP4 Axis, Heme Oxygenase, and Chemokines in MSC-Mφ Interaction

Several secreted factors were shown to play key roles in MSC/Mφ interplay, including cytokines and chemokines that are important mediators in inflammation [117].

##### Cytokines

IL-6, produced by MSCs and present in the inflammatory environment, is a very important regulator of Mφ polarization toward an IL-10-producing M2-like phenotype. This polarization is initiated by MSC/Mφ cell–cell contact and is also dependent on other factors secreted by MSCs, including the antibacterial molecule indoleamine (INDO) [136] and keratinocyte growth factor (KGF) [47]. The ability of MSCs to favor the emergence of CD4^+^CD25^+^FoxP3^+^ regulatory T cells (T-regs) [137] also affects Mφ polarization. Importantly, activated MSCs secrete TSG-6 that interacts with CD44 on resident Mφs and decreases the secretion of pro-inflammatory factors by reducing TLR2/NFκ-B signaling [118,138,139].

##### COX/PGE2/EP4 Axis

The secretion of prostaglandin (PG)E2 by MSCs serves as a crucial regulator of MSC/Mφ interaction controlling inflammation and tissue homeostasis, repair, and regeneration [124].

PGE2 is derived from arachidonic acid by cyclooxygenase synthases (constitutively active COX1 and inducible COX2) and PGE synthases [140]. The studies by Németh and colleagues [123] have revealed that BM-MSCs, especially if activated by LPS or TNF-α, release PGE2 that acts on the Mφs through PG receptors (EP2 and EP4) triggering IL-10 secretion. The role of PGE2 and the mechanism involving IL-6 and IL-10 was later confirmed and elaborated by many other groups in a variety of injury and disease models [122,133,141,142,143].

Heat shock factor 1 (HSF1) translocation into the nucleus of MSCs was also shown to induce the COX2/PGE2 pathway and MSC-directed immune suppression [56]. Moreover, Nox-2-dependent ROS production was crucial for Mφ bacterial killing and dependent on PGE2 and phosphatidylinositol 3 (PI3)-kinase [97].

##### Growth Factors and HO-1

TGF-β secreted by MSCs skewed LPS-stimulated Mφ polarization towards the M2-like phenotype and improved Mφ phagocytic ability via the Akt/FoxO1 pathway [5].

Naïve UC-MSCs increase HO-1 expression and phagocytic capabilities in hMφs. This effect is augmented by transfection of MSCs with IL-10 and is abolished by PGE2 and lipoxygenase A4 blockade. This mechanism was also verified in a co-culture of MSCs and Mφs isolated from rat lungs with induced *E. coli* pneumosepsis [58] and from the peritoneal cavity of septic rats [34].

Vascular growth factor (VEGF) and angiopoietin-1 (Ang-1), secreted by MSCs or present on MSC-EVs in the form of mRNA, have been found to be important not only for the restoration of vascular stability but also for the repair of acute lung injury [109] or resolution of allergic asthma in mice [108] through Mφ immunomodulation.

##### Chemokines

Chemokines also play a prominent role as a nexus between MSCs and Mφs [66]. MSC-derived CCL-2 acts as Mφ and monocyte chemoattractant protein (MCP-1). This chemokine forms a heterodimer with MSC-derived CXCL-12 and triggers Mφ IL-10 production and M2-like Mφ polarization. Consequently, CCL-2-null MSCs lose their anti-inflammatory potential [83]. Furthermore, IFN-γ upregulates CCL2 expression in MSCs; therefore, CCL2-deficient MSCs were ineffective when administered into IFN-γ- or IFN-γ receptor-deficient recipients or failed to suppress allergen-induced lung inflammation [77]. Other chemokines in tandem with CCL-2 could increase Mφ anti-inflammatory potential including stromal-derived factor-1 (SDF-1 or CXCL12) [105,144], CCL4 (MIP-1β), and CCL5 (RANTES) [145]. The role of these chemokines is summarized in a comprehensive review by Galipeau [66].

#### 4.2.2. Role of MSC-Derived Extracellular Vesicles (EVs), mRNA, MicroRNA, and Mitochondrial Transfer in Immunomodulation through Mφs

MSCs have been known to release EVs that contain a variety of cargos including endosomal and plasma membrane, intracellular organelles (e.g., mitochondria—Mt), cytokines, growth factors, signaling lipids, mRNAs, and regulatory miRNAs [146]. EVs released by MSCs might be as effective in therapy as whole MSCs. Thus, MSC-EVs represent an appealing option for cell-free regenerative medicine [147] as their content could be delivered to immune cells present in an inflammatory environment (M1-like Mφs, dendritic, CD4+Th1, and Th17 cells), promoting their phenotypic switch into immunosuppressive M2-like Mφs, tolerogenic DCs, and regulatory T cells [91]. Moreover, EVs could be exploited as attractive tools for diagnostic and therapeutic agent delivery [148]. A meta-analysis screening 52 articles [149] demonstrates the clear potential of MSC-EVs as a therapeutic tool for acute and chronic lung diseases in particular. MSC-EVs or exosomes are already being used in a few dozen clinical trials in [150] and [151], including in some COVID-19 pneumonia trials.

Several studies demonstrated that the mechanism of the beneficial effects of MSC-EVs are executed, at least in part, through their interaction with Mφ.

An in vitro co-culture of MSCs with Mφs enhanced their M2-like polarization mainly through MSC-derived exosomes [126,152]. Infusing MSCs lacking exosomes led to a lower number of M2-like Mφs in vivo. Further, the hyperoxia-induced reduction in CD163-positive Mφs was prevented by MSC-EVs in a rat model of bronchopulmonary dysplasia [64]. In an ALI *E. coli* endotoxin mouse model, instilled MSC-EVs reduced pulmonary edema and lung inflammation by decreasing Mφ inflammatory protein-2 levels in the BAL fluid [153].

A co-culture of MSCs or MSC-EVs with regulatory Mφs (M2b subset) amplified the pro-resolving properties of Mφs [154], while EVs from (IFN)-γ-primed MSCs were more efficient than ones from naïve MSCs in enhancing Mφ phagocytosis [53].

Exciting data suggest that mitochondrial transfer from MSC-EVs to recipient cells, including Mφs, is an important mechanism for enhancing Mφ anti-inflammatory and regenerative capacity in injury and inflammation. Using a co-culture system consisting of MSCs and Mφs, Yuan et al. [155] showed that MSC-derived Mt were transferred into Mφs, which contributed to their M2 polarization. The beneficial outcome of ALI in mice subjected to the adoptive transfer of AMφs pretreated with MSC-EVs was dependent on EV-mediated mitochondrial transfer [44].

In addition to the above-described cargo, different MSC-MV-derived microRNAs were also found to be important mediators in Mφ switching towards an M2-like phenotype, including miR-223 [126] and miR-182 [152]. The alleviation of ARDS in mice was found to be mediated by miR181 [156] while miR-466 contributed to MSC-EV-induced improvement in a multidrug-resistant pseudomonas aeruginosa pneumonia mouse model [157].

Combined, these studies provide evidence that EVs and their cargo are indeed key mediators of the effects of MSCs. An improved understanding of the role of the EV cargo will be a key aspect of improving the design of MSC therapy.

### 4.3. Role of Autophagy, Mitophagy, and Oxidative Stress in MSC-Mφ Interplay

Autophagy, the main cellular mechanism for degrading and recycling intracellular proteins and organelles, plays an important role in maintaining bioenergetic homeostasis in health and disease and may contribute to the therapeutic action of MSCs [158]. The modulation of autophagy may also change Mφ efficiency and polarization. Indeed, BM-MSCs were shown to exert beneficial effects in a mouse model of sepsis acting primarily by enhancing mitophagy in Mφs and decreasing mitochondrial ROS, thus inhibiting NLRP3 inflammasome activation [159]. The activation of autophagy, HO-1, and mitochondrial biogenesis occurs after MSC exposure to Mt isolated from somatic cells. During tissue injury, MSCs are prompted by Mt released from damaged cells to donate Mt to injured cells, thereby enhancing tissue reparation [129].

Oxidative stress, frequently present in inflamed and damaged tissue, is also known to activate autophagic processes [160,161]. BM-MSCs are able to modulate autophagy in Mφs through the PI3K/Akt/HO-1 signaling pathway [162] and protect rats against liver I/R injury via the promotion of HO-1-mediated autophagy [136].

In an MSC-Mφ co-culture system and mouse model of silicosis [163], MSCs reduced intracellular oxidative stress. This was attributed to targeting depolarized Mt and releasing MVs containing entire Mt selected for mitophagy, lysosomes, and several miRNAs (especially miR451a). The vesicles formed by this process are then engulfed and reutilized by Mφs causing the repression of TLR and NF-κB signaling in Mφs and decreasing the production of inflammatory and pro-fibrotic mediators.

MSC/Mφ bidirectional communication plays a pivotal role in promoting the regeneration and recovery of injured lungs. In the environment of damaged lungs, Mφs are activated (M1-like phenotype), secreting, among others, pro-inflammatory cytokines (TNF-α, IL-1α, IL-1β, IL-6) [92,93], chemokines (CCL2, CCL8, CXCL-9, CXCL-10) [66,90,92], ROS, and NO [97,98,123]. These factors trigger the COX-2 pathway in MSCs which produces PGE2 [97,123,124], resolvins (Rv-D1, Rv-E1, Rv-E2), protectins [125], and lipoxins (LXA4) [34] which are actively involved in lung reparation [164]. These factors also trigger IL-10 secretion from M1-like Mφs, enhance their phagocytic properties, and induce the switch towards M2-like anti-inflammatory Mφs [90,93]. M2-like Mφs are characterized by the expression of CD163, CD206, and FIZZ1 [92]. CD200 [84] and TGS-6 [118] secreted by MSCs act on Mφ receptors (CD44 and CD200R) promoting polarization to the M2-like phenotype. They also inhibit T cell proliferation favoring T-regs that assist in restoring immune homeostasis. Anti-inflammatory molecules KGF [47,52,105,116] and INDO [107] secreted by MSCs have direct antibacterial effects but also promote Mφ activation and phagocytosis [165] and regulate Mφ recruitment and polarization [115,166]. Direct contact between MSCs and Mφs [84] and nanotube formation allow the direct transfer of MSC-derived MVs [146,163] and Mt from the MSCs to Mφs to enhance Mφ phagocytosis [119]. EVs released by MSCs [53,91,112,113,126,127,130,131] and their cargo of proteins, miRNA [51,157], and cellular compartments, including Mt [155], might be as effective as whole MSCs as therapeutics.

## 5. Therapeutic Potential of MSC-Mφ Interaction and Lung Injury Resolution

The MSC-Mφ interplay has a pivotal role in lung injury combating inflammation and promoting injury resolution [66,71,72,95,111,167]. For tissue recovery, crucial effects include the dampening of inflammation, release of IL-10 by Mφs, and favourable effects of eicosapentaenoic acid (EPA) and its derivatives, resolvins (Rv- D1, E1, and E2) and protectins secreted by MSCs [125]. EPA also gives rise to eicosanoids and their metabolites PGE2 and lipoxin A4 in MSCs [58,97,123]. These are key for IL-10 induction and enhanced Mφ phagocytosis, contributing to the resolution of inflammation and tissue recovery. EPA pre-conditioning of MSCs further reduces sepsis-induced lung injury and leads to faster recovery [125].

### 5.1. Reparatory Potential of MSC-Mφ Interaction in Chronic Lung Diseases

The reparative effect of MSCs and MSC-Mφ interaction allowed their use not only in acute but also in chronic lung injury [61,62,149,168,169], as mentioned before. MSCs or their secreted factors and EVs were found to be beneficial in many models of chronic lung injury including in allergic, ragweed, or ovalbumin-induced asthma models [55,170,171]. In asthmatic mice, TGF-β production by MSCs increased the presence of regulatory Mφs and T cells, which helped restore cytokine balance and prevented harmful allergic responses [172,173]. In bronchopulmonary dysplasia (BPD) mouse models, MSC-EVs reduced lung injuries partly by increasing M2-like interstitial/alveolar Mφ polarization and their anti-inflammatory and anti-proliferative action [64,130]. In COPD, MSCs act by attenuating the airway infiltration of neutrophils and Mφs, leading to decreased production of IL-1β and IL-6 while increasing IL-10 and the levels of growth factors (VEGF, HGF, EGF, TGF-β), therefore boosting tissue repair [63]. Similarly, in models for idiopathic pulmonary fibrosis (IPF), MSCs or MSC-derived EVs could attenuate lung fibrosis by acting on Mφs by promoting ATII cell proliferation and by inhibiting lung fibroblast proliferation [65,131]. MSC-derived EVs were also able to prevent or reverse lung fibrosis in bleomycin-treated mice by modulating pulmonary Mφ phenotypes, shifting them to an immunoregulatory and anti-inflammatory phenotype [131].

However, collecting more data on the mechanisms of MSC action and MSC-Mφ interaction in clinical studies and particularly in chronic lung disease is imperative. Randomized controlled trials with large cohorts of patients and with a mechanistic approach are needed. Clinical trials in IPF, asthma, silicosis, and COPD conducted so far are mainly oriented to assessing the safety of the treatment, with single or two MSC doses and mostly focused on the short-term effects of therapy [62,174,175,176,177].

Despite the encouraging results from these studies, special attention needs to be paid to the possibility of worsening lung function and fibrosis caused by the administration of MSC in the chronic stage of IPF [178].

### 5.2. Clinical Studies

MSCs are being tested in clinical trials in critically ill patients with sepsis, ARDS, COVID-19 pneumonia, ARDS with malignances, pulmonary emphysema, pulmonary arterial hypertension, aplastic anemia, cancer, and liver cirrhosis, as well as for the prevention of graft vs. host disease (GVHD), etc. The treatments assessed involve autologous or allogeneic BM-MSCs, umbilical cord MSCs, or exosomes derived from MSCs. Moreover, allogeneic MSCs are also safely used in some clinical studies including trials for pulmonary emphysema and patients with COVID-19 ARDS.

These clinical studies use MSCs or MSC-EVs in lung injury treatment [179], most frequently in ARDS [80,180,181] including SARS-CoV-2 infection (Table 3).

### 5.3. MSC-Mφ Interaction in COVID-19—More Studies Are Needed

In addition to the above described effects, antiviral properties have also been attributed to MSCs and MSC-EVs. MSCs could inhibit virus replication and shedding through IDO and LL37 secretion and improve influenza-induced viral pneumonia and ARDS due to anti-inflammatory and reparative potential, partly through the interaction with Mφ [182,183].

SARS-CoV2 enters cells, including alveolar epithelial and lung capillary endothelial cells, through the widely expressed angiotensin-converting enzyme 2 (ACE2) receptor [184]. MSCs are ACE2-negative, and therefore they are resistant to SARS-CoV-2 infection and retain their immunomodulatory activities when encountering the virus [185,186]. This important feature favors MSC therapeutic use in COVID-19 [187]. Interaction with innate immune cells, especially Mφs, and molecules secreted by MSCs and Mφs, described above, could dampen immune hyper-activation in SARS-CoV-2 infection. Further, MSCs also have anti-fibrotic and lung reparatory effects [188]. Thus, MSCs or MSC-derived EVs could be used in a potential supportive and/or curative strategy for COVID-19, especially in critically ill patients [127,186,189]. So far, dozens of clinical trials have been performed in COVID-19 patients with severe disease [190]. Since most of these are still ongoing, limited preliminary data are available [189]. The results so far from studies registered on the US ClinicalTrials.gov page (Table 3) are promising, showing the absence of adverse effects in COVID-19 patients treated with MSCs or MSC-EVs [191,192,193], reduced expression of pro-inflammatory cytokines [194], and improved recovery time [195]. Some studies from non-US goverment pages (China, Iran) showed an increase in IL-10 expression with improved lung function outcomes [196] and, mortality, as well as positive effects on chest imaging results [197] and oxygen saturation in the MSC-treated COVID-19 group [195,198].

**Table 3 ijms-24-03376-t003:** Randomized controlled trials of MSC therapy in severe ARDS (including ARDS caused by COVID-19).

Study Type/Patient Cohort	Intervention	Outcomes Measured and Results	Reference/Trial Number
Phase 1 moderate–severe ARDS 12 patients (pts).	Adipose MSCs—allogeneic 1x i.v. -1 million cells/kg or placebo.	No cell toxicity or SAEs. No improvement in length of hospital stay or ventilator-free days or change in biomarkers.	[199] Zheng et al., 2014 NCT01902082
Phase 1 (STAR) 9 (pts.); moderate-to-severe ARDS.	BM-MSCs—allogeneic 1x i.v.: 1, 5, or 10 million cells/kg (3 pt./each dose).	Safety trial: single dose of allogeneic BM-MSCs was safe and well tolerated.	[180] Wilson et al., 2015 NCT01775774
Phase 2a (STAR) Moderate–severe ARDS 60 ventilated pts.	BM-MSCs allogeneic—1x i.v.: 2:1 either 10 million/kg of MSCs or placebo.	28-day mortality did not differ after adjustments for APACHE III score.	[80] Matthay et al., 2019 NCT02097641
Nested cohort within a phase 2a (STAR); moderate–severe ARDS 27 pts.	BM-MSCs—allogeneic 1x i.v.: 10 million/kg, n = 17 pts, and placebo, n = 10 pts.	MSC treatment significantly reduced airspace total protein, Ang-2, IL-6, and soluble TNF receptor-1 concentrations.	[181] Wick et al., 2021 NCT02097641
Phase 1/2a COVID-19 ARDS 24 pts (1:1).	UC-MSCs—2x i.v. (100 million cells/infusion) + heparin; placebo vehicle + heparin.	No AEs and SAEs with cell treatment; improvement of patient survival and time to recovery.	[195] Lanzoni et al., 2021 NCT04355728
Phase 1 COVID-19 with mild–severe ARDS (REALIST) 9 pts.	UC-MSCs-CD362 (Syndecan-2) enriched (ORBCEL-3)—1x i.v.: 100, 200, or 400 million cells/infusion (3 pts/each dose).	Well tolerated and no dose-limiting toxicity. Safe to proceed to Phase 2 trial.	[192] Gorman et al., 2021 NCT03042143
Phase 1/2a COVID-19; critically ill 40 pts (1:1).	UC-MSCs + standard care (Oseltamivir and Azithromycin)—1x i.v. 1 million cells/kg or placebo + standard care.	Improved survival rate, no changes in ICU stay or ventilator use, and no AEs. IL-6 reduced	[194] Dilogo et al., 2021 NCT04457609
Phase 2 COVID-19 with severe ARDS 100 pts, (2:1).	UC-MSCs—3x i.v. (40 million cells/infusion) or placebo.	Improvement in whole-lung lesion volume and no difference in SAEs.	[191] Shi et al., 2021 NCT04288102
Phase2b COVID-19 with mild–severe ARDS 45 pts.	UC-MSCs—3x i.v. 1 million/kg = 21 pts, or placebo = 24 pts.	No SAEs associated with repeated cell infusions. PaO_2_/FiO_2_ changes did not differ between the groups.	[193] Monsel et al., 2022 NCT04333368

## 6. Discussion and Conclusions

The overall objective of this review was to provide an overview of up-to-date knowledge of MSC-Mφ crosstalk and its importance in acute and chronic pulmonary disease treatment and resolution of lung injury. The beneficial effects are thought to be due to bidirectional MSC-Mφ communication, which is attributed to direct contact, soluble factor secretion/activation, and organelle transfer. Further, MSC-derived EVs represent an appealing option for cell-free regenerative medicine. Their delivery in a pro-inflammatory environment dominated by M1-like Mφs could promote Mφ transition towards immunosuppressive and pro-reparatory M2-like cells. Moreover, EVs could be used as an attractive tool for diagnostic and therapeutic agent delivery.

It is important to note that the outcome of the MSC-Mφ interaction depends on many factors, and the final outcome is not always beneficial. The environment, disease stage, and other components of innate and adaptive immunity could all influence the MSC-Mφ interplay and its consequences. More research is clearly needed to define these factors in specific diseases and disease stages and to make this interaction predictable and modifiable, assuring favorable outcomes.

Importantly, the clinical use of MSCs is still restricted, and clinical trials conducted so far have mainly investigated the safety of MSC-based therapy in acute and chronic lung diseases and in patients with diseases resistant to other therapeutic options. This clearly precludes the immediate use of MSCs in lung diseases, even though novel therapeutic approaches are desperately needed. More critically, there is no effective treatment for ARDS, a condition that develops and progresses rapidly and has high mortality [200]. For this reason, most of the clinical studies exploring the use of MSCs or their derivatives are conducted in ARDS patients [200,201]. However, these studies were carried out with a relatively small number of enrolled patients. Therefore, large-scale clinical trials are lacking. The manipulation of MSC-Mφ interaction described in this review represents an attractive option for the improvement of MSC efficacy in the treatment of lung diseases. However, more detailed studies are needed. These should include cytokine measurements and the isolation of monocytes and Mφ from plasma, BALF, and patient lung biopsies to characterize Mφ phenotype, phagocytic properties, and secretion profiles. Moreover, since the effect of MSCs depends on the microenvironment, a better understanding of the composition of the microenvironment in lung diseases is paramount for developing appropriate strategies to control MSC action and MSC-Mφ crosstalk. Our increasing knowledge and understanding of the mechanisms of action of MSCs and the MSC-Mφ interaction in particular, together with use of cell-free MSC derivatives (conditioned media, EVs, Mt), could help develop strategies for the effective treatment of lung injury. Ideally, this should happen in the imminent future as there is a substantial unmet medical need for efficacious treatment options.

## Figures and Tables

**Figure 1 ijms-24-03376-f001:**
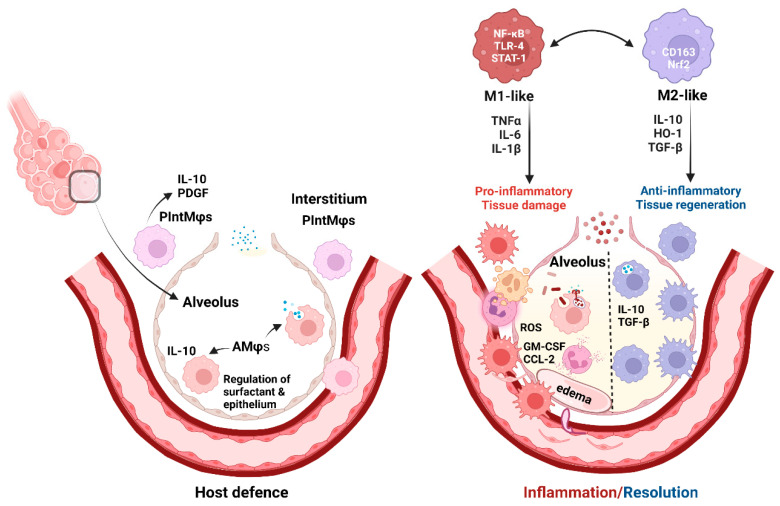
Pulmonary macrophages in host defense and inflammation/resolution (created with BioRender.com).

**Figure 2 ijms-24-03376-f002:**
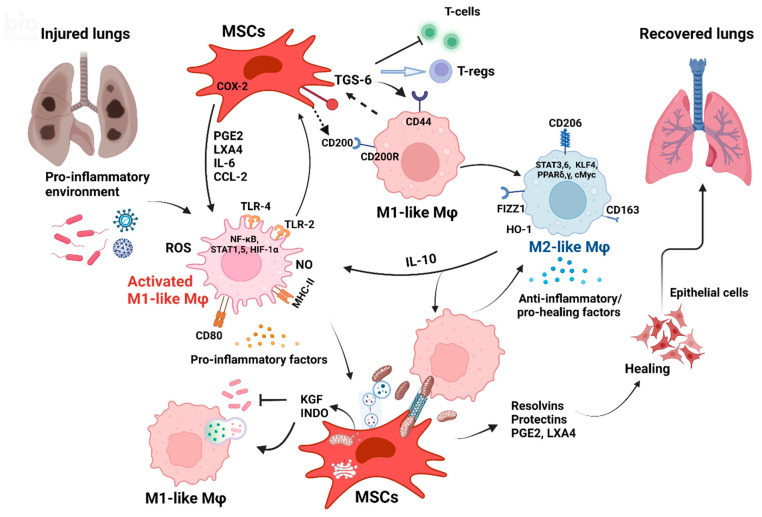
Crosstalk between MSCs and Mφs is crucial for lung injury recovery. (Created with BioRender.com.)

**Table 2 ijms-24-03376-t002:** Key factors secreted by M1-like and M2-like Mφs and MSCs alone and the effects of MSC-Mφ interaction on secreted factors.

M1-like Mφs	M2-like Mφs	MSCs	MSC-Mφ Interaction	
			MSCs *	Mφ
IL-1α, Il-1β [92,93] IL-6, 12, 23 [69,92,94] TNF-α [92,93,94] CCL-2, 8, 10, 15, 19, 20 [66,90,92] CXCL-9, 10, 11, 16, 17 [95,96] ROS (by Nox-2), NO (by iNOS), [92,94,97,98]	IL-4,10, 13 [77,92,93] CCL-1, 17, 18, 22, 24 [92,95,99] CXCL-13 [100] CXCL-12 [101] TGF-β [94,95,102] SDF-1 and VEGF [103,104] Arg-1 [23,92] HO-1 [98]	CCL-1-2, 4-5 [66] CXCL-8, 10, 12 [66,105] KGF [47,105] INDO [106], TSG-6 [107] Ang-1, VEGF, HGF, IGF-1 [105,108,109,110] EVs, Mt, MiRs [53,111,112,113]	CCL-2, 5, 7 [66,114] CXCL-8-12 [66] KGF [52,105,115,116] NO, TGFβ [34,117] INDO, TSG-6 [107,118,119] Ang-1, VEGF, HGF, IGF-1 [108,109,110,120,121] COX-2/PGE2, LXA4 [34,97,122,123,124] Rv-D1, Rv-E1, E2, Protectins [125] EVs, Mt, MiRs [51,53,91,113,126,127,128,129,130,131,132]	↑ M2 Mφ and their ILs (by STAT 3,6, PPARγδ [30,56,95] ↓ M1 Mφ and their ILs [109,133] HO-1, antioxidants [34,134] IL-10 [32,123] TGF-β [32,95,132] Arg-1 [92,135]

* MSCs secrete mostly the same factors with or without Mφs; however, interaction with Mφs increases the quantity of these secreted factors; ↑ upregulation; ↓ downregulation.

## Data Availability

Not applicable.

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
