# Peer review of "Key Role of Mesenchymal Stromal Cell Interaction with Macrophages in Promoting Repair of Lung Injury"

_ijms, 2023, doi:10.3390/ijms24043376_

Round 1

Reviewer 1 Report

I recommend that this paper be published in the journal because the contents represent an important aspect. Here are some suggestions:

1: What motivated the authors to prepare the article. An “Introduction” should be added in the revised version to highlight the novelty of this work clearly.

2: “1. The role of lung tissue macrophages in the pathogenesis of lung injury” and “3. Crosstalk between MSCs and Mφs – mechanisms of action ” of the manuscript should be more concise. Authors should thoroughly improve these parts.

3: The manuscript has too many sections, please check and modify.

4: The authors should enrich the “Discussion” part in the revised version. The authors should add the significance of this research and its drawbacks and challenges.

5: Abbreviations should be placed at the end of the manuscript.

Author Response

I recommend that this paper be published in the journal because the contents represent an important aspect. Here are some suggestions:

1: What motivated the authors to prepare the article. An “Introduction” should be added in the revised version to highlight the novelty of this work clearly.

Response: The new Introduction section has been added to highlight novel aspects and the importance of this work.

2: “1. The role of lung tissue macrophages in the pathogenesis of lung injury” and “3. Crosstalk between MSCs and Mφs – mechanisms of “action” of the manuscript should be more concise. Authors should thoroughly improve these parts.

Response: We have significantly shortened the text in these sections and merged some of the subsections.

3: The manuscript has too many sections, please check and modify.

Response: Following the reviewer’s recommendation, we significantly reduced the number of sub-sections; for example, we merged and shortened several subsections in section 2 (e.g. merging and shortening sections 2.1 and 2.2.)

4: The authors should enrich the “Discussion” part in the revised version. The authors should add the significance of this research and its drawbacks and challenges.

Response: The new section contains more detailed discussion of the drawbacks and challenges, as suggested by the reviewer.

5: Abbreviations should be placed at the end of the manuscript.

Response: Done

Reviewer 2 Report

The manuscript entitled "Key Role of Mesenchymal Stromal Cell Interaction with Macrophages in Promoting Repair of Lung Injury" authored by Jerkic et al. present an overview of the ways in which MSCs interact with other key cells such as macrophages in aiding the lung repair and regeneration. Authors neatly described the role of lung tissue macrophages in the pathogenesis of lung injury, followed by various studies on MSCs in lung injury treatment, and then focused on key mechanisms of interactions between MSCs and macrophages. Preclinical and clinical case studies showing the therapeutic potential of MSCs and macrophage interaction and lung injury resolution were also well-described. Overall, the article was well-structured, focused and comprehensive.

Author Response

We thank the Reviewer for such a positive opinion about our Review paper.

Reviewer 3 Report

Please check the attached document

Author Response

The review by Jerkic et al entitled ‘Key Role of Mesenchymal Stromal Cell Interacting with Macrophages in Promoting Repair of Lung Injury' is a well written review that covers the topic in appropriate detail and provides a balanced summary about the study of Macs-MSCs in preclinical and clinical level. I have only a few minor suggestions to the authors to improve the manuscript.

Response: We thank Reviewer for such a positive opinion about our Review paper.

Both figures covered extensive details about the inflammation, cytokine/chemokine secretion etc, which is difficult to read for readers. I would suggest the author to simplify the figure details, particular for Figure 2. Too many arrows and cytokine listed in the figures. If possible, the author could reorganize the structure of figures and show the interaction of Macs and MSCs through flow-chart with steps. Also, please modify both figure legends, and descries the figures based on the information covered in figures.

Response: We have modified/simplified Figure 1 and 2. In Figure 2 the names of the cytokines were removed, one of the depicted MSCs was removed and the information merged into other parts of the figure and some arrows were also removed. We have also assured that the Figure legend describes exactly what it is depicted in the Figure. We would like to point out that the figure depicts complex multidirectional effects with numerous feedback that we thought would be best represented in a figure rather than a flow-chart.

Table 2 is confusing, what are the difference between each row listed there? Any rational the authors group the cytokines in that way? The author may add a column to describe that.

Response: We agree with the Reviewer that the original table was not clear. The new reorganized table has no rows, and each column represents a cell type as indicated above the column.

Line 68, Line 267, the format of title is different. Please check the format title at different level across the manuscript, and make sure they are consistent, and in right format.

Response: We thank the Reviewer for pointing out this formatting error. The format of the title and subtitle have been corrected. We also corrected the errors or typos listed below.

Line 43, missing a period at the end of sentence. 

Line 187, ‘;’ should be ‘)’ 

Line 439, ‘revers’ should be ‘reverse’. 

Line 480, Table 3 should in bold. 

Response: Thank you, they are all corrected.

Round 2

Reviewer 1 Report

The authors have addressed my comments to my satisfaction. I recommend it for publication in its current form.

Reviewer 3 Report

Thank you for addressing my comments, I accepted them all, and the manuscript looks much better.